# Clinical Utility of Prostate Health Index for Diagnosis of Prostate Cancer in Patients with PI-RADS 3 Lesions

**DOI:** 10.3390/cancers14174174

**Published:** 2022-08-29

**Authors:** Chung-Un Lee, Sang-Min Lee, Jae-Hoon Chung, Minyong Kang, Hyun-Hwan Sung, Hwang-Gyun Jeon, Byong-Chang Jeong, Seong-Il Seo, Seong-Soo Jeon, Hyun-Moo Lee, Wan Song

**Affiliations:** Department of Urology, Samsung Medical Center, Sungkyunkwan University School of Medicine, Seoul 06351, Korea

**Keywords:** prostate cancer, prostate imaging reporting and data system, prostate health index, multiparametric magnetic resonance imaging

## Abstract

**Simple Summary:**

Multi-parametric magnetic resonance imaging (mpMRI) is regarded as an essential tool for identifying prostate cancer (PCa) in suspected cases. However, unnecessary biopsies continue to be performed in real clinics, especially for prostate imaging reporting and data system version 2 (PI-RADS v2) score-3 lesions, corresponding to the “gray zone”. To aid the diagnosis of PCa, as well as of clinically significant PCa (csPCa), in patients with PI-RADSv2 score-3 lesions, we evaluated the clinical utility of the prostate health index (PHI). When a biopsy was restricted to those patients with PI-RADSv2 score-3 lesions and a PHI of ≥30, 34.4% of unnecessary biopsies could be avoided at the cost of missing 8.3% of overall PCa cases. However, there were no cases of missed csPCa diagnosis. The combination of PHI and PI-RADSv2 score-3 lesions offered higher accuracy in the diagnosis of PCa as well as of csPCa.

**Abstract:**

The risk of prostate cancer (PCa) in prostate imaging reporting and data system version 2 (PI-RADSv2) score-3 lesions is equivocal; it is regarded as an intermediate status of presented PCa. In this study, we evaluated the clinical utility of the prostate health index (PHI) for the diagnosis of PCa and clinically significant PCa (csPCa) in patients with PI-RADSv2 score-3 lesions. The study cohort included patients who underwent a transrectal ultrasound (TRUS)-guided, cognitive-targeted biopsy for PI-RADSv2 score-3 lesions between November 2018 and April 2021. Before prostate biopsy, the prostate-specific antigen (PSA) derivatives, such as total PSA (tPSA), [-2] proPSA (p2PSA) and free PSA (fPSA) were determined. The calculation equation of PHI is as follows: [(p2PSA/fPSA) × tPSA ½]. Using a receiver operating characteristic (ROC) curve analysis, the values of PSA derivatives measured by the area under the ROC curve (AUC) were compared. For this study, csPCa was defined as Gleason grade 2 or higher. Of the 392 patients with PI-RADSv2 score-3 lesions, PCa was confirmed in 121 (30.9%) patients, including 59 (15.1%) confirmed to have csPCa. Of all the PSA derivatives, PHI and PSA density (PSAD) showed better performance in predicting overall PCa and csPCa, compared with PSA (all *p* < 0.05). The AUC of the PHI for predicting overall PCa and csPCa were 0.807 (95% confidence interval (CI): 0.710–0.906, *p* = 0.001) and 0.819 (95% CI: 0.723–0.922, *p* < 0.001), respectively. By the threshold of 30, PHI was 91.7% sensitive and 46.1% specific for overall PCa, and was 100% sensitive for csPCa. Using 30 as a threshold for PHI, 34.4% of unnecessary biopsies could have been avoided, at the cost of 8.3% of overall PCa, but would include all csPCa.

## 1. Introduction

Prostate cancer (PCa) is known to be the second most commonly diagnosed cancer worldwide and was the most diagnosed cancer in men in 106 countries in 2018. [1] Traditionally, the serum prostate-specific antigen (PSA) has been used as a biomarker of prostate cancer. The early detection of PCa is possible with the widely performed PSA screening test; however, many unnecessary prostate biopsies are still performed in men with a PSA level that is reported in the range of 2.5–10 ng/mL, the so-called “gray zone,” which can be attributed to the high sensitivity and low specificity of PSA toward PCa diagnosis. Several diagnostic tools have been developed to improve the detection of PCa and have been studied for their clinical efficacy. For example, free PSA (fPSA), free-to-total PSA ratio, PSA density, and PSA velocity have been introduced to compensate for the shortcomings of PSA. In addition, the PSA isoform and derivatives are reported to be more accurate in diagnosing PCa. [2] However, these diagnostic tools have not proven to be a substitute for PSA. In the past decade, the prostate health index (PHI) has been introduced, which is calculated by using the total PSA (tPSA), fPSA, and a PSA isoform [-2]proPSA (p2PSA), and it outperformed tPSA or fPSA in the diagnosis of PCa and clinically significant PCa (csPCa) in the initial biopsy, as well as in a repeat biopsy [3,4,5,6,7].

Recently, multi-parametric magnetic resonance imaging (mpMRI) was introduced and is regarded as an essential tool for identifying PCa in suspected cases [8,9]. Therefore, major guidelines, including the European Association of Urology (EAU) and the American Urology Association (AUA), recommend that clinicians should perform an mpMRI before performing a biopsy in patients who are either biopsy-naïve or who have had negative biopsy reports [10,11]. An international collaboration with the European Society of Uro-Radiology (ESUR) and the American College of Radiology (ACR) has released the prostate imaging reporting and data system, version 2 (PI-RADSv2), in 2015, and the system is widely used as a standard reporting system of mpMRI [12].

The use of these diagnostic tools can avoid unnecessary prostate biopsies in patients, and can thereby aid in the diagnosis of csPCa [13,14,15]. However, unnecessary biopsies continue to be performed in real clinics, especially for PI-RADS v2 score-3 lesions corresponding to the “gray zone” of the PI-RADS v2 scoring system. Owing to the clinicians’ concerns regarding performing biopsies of these lesions, the need for other biomarkers that may help to determine the diagnosis of such lesions is widely recognized. In this study, we evaluated the clinical utility of PHI for aiding the diagnosis of PCa, as well as csPCa, in patients with PI-RADSv2 score-3 lesions.

## 2. Materials and Methods

### 2.1. Study Population

The Institutional Review Board of the Samsung Medical Center (IRB No. 2021-05-103) has approved this study; informed consent was waived by the IRB, due to the retrospective nature of the study. Study protocols were strictly adhered to, following the principles of the Declaration of Helsinki.

We retrospectively reviewed 466 patients whose PSA was 2.5–10.0 ng/mL, and PI-RADS v2 score-3 lesions were confirmed by an mpMRI, between November 2018 and April 2021. From the cohort, 425 patients underwent transrectal ultrasound (TRUS)-guided, cognitive-targeted biopsy for PI-RADSv2 score-3 lesions; 41 patients were excluded because of incomplete findings in terms of the target lesions. In addition, 33 patients were also excluded because of incomplete clinical data; finally, 392 patients were enrolled in our study (Figure 1).

### 2.2. Laboratory Tests

All patients’ serum samples were collected before performing a TRUS-guided biopsy (TRUS-Bx). The samples were evaluated for specific PSA parameters, including tPSA, fPSA, and p2PSA. Thereafter, the blood samples were centrifuged for three hours and frozen at −20 to −80 ℃ until analysis. We used an immunoassay system (Beckman Coulter DxI 800 Immunoassay System) to determine PHI, which was calculated using the following formula: [16].
PHI = (p2PSA/fPSA) × (tPSA ½) 

### 2.3. mpMRI Protocol and Interpretation

The mpMRI was conducted with a 3.0-T MRI instrument (Intera Achieva TX, Philips Healthcare, Best, the Netherlands), which includes a 6-channel phase-array body coil, and was performed before TRUS-bx. 

The mpMRI’s scanning protocol included T1- and T2-weighted imaging, diffusion-weighted imaging with b values of 0, 100, 1000, and 1500 s/mm^2^, and dynamic contrast-enhanced imaging after the injection of a gadolinium diethylenetriamine penta-acetic acid (Gadovist, Schering), as per the ESUR guidelines [17]. 

The picture archiving and communication system (Centricity, GE Healthcare, Barrington, IL, USA) was used to upload the magnetic resonance images, which were interpreted by two uro-radiologists with more than 10 years of experience with prostate MRI interpretation. The reviewers scored an index lesion according to the PI-RADS v2 system, using a 5-point scale [18].

### 2.4. Biopsy Protocol

As prostate biopsy may lead to PSA elevation and the misinterpretation of mpMRI results due to hemorrhage, a laboratory examination, and mpMRI were performed before the prostate biopsy. In addition, prostate volume was assessed by TRUS, applying the ellipsoid formula (L × W × H × 0.523) before performing the prostate biopsy. All prostate biopsies were performed following the technique of a conventional TRUS-Bx under local anesthesia. Initially, cognitive-targeted biopsy with more than one core sample from the target area was performed, depending on the mpMRI results; additionally, 12-core specimens were obtained thereafter with an 18-gauge core biopsy needle mounted on an automatic biopsy gun, using a biplane TRUS probe (BK Medical, Herlev, DenmarkTransducer 8818).

### 2.5. Histopathological Analysis

The biopsy specimens were reviewed by an experienced uro-pathologist. The PCa was evaluated according to the 2014 International Society of Urological Pathology (ISUP) Consensus Conference guidelines [19]. Specifically, grade group (GG) 1 is equivalent to a Gleason score (GS) of ≤3 + 3; GG 2 equals GS 3 + 4; GG 3 is equivalent to GS 4 + 3; GG 4 is equivalent to GS 8; GG 5 is equivalent to GS 9 − 10. We defined csPCa as GG 2 or higher.

### 2.6. Statistical Analysis

The continuous data in this paper were shown as median (interquartile range, IQR) and mean (standard deviation, SD); fixed data that were arranged as categorical data were presented as an absolute value (percentage). The normality of the continuous data was evaluated using the Kolmogorov–Smirnov test. Continuous variables were quantified and analyzed via an independent Student’s *t*-test; Pearson’s chi-squared test and Fisher’s exact test were used to analyze the categorical variables between PCa and the negative results at biopsy. The sensitivity, specificity, and positive and negative predictive values (PPV and NPV) of PHI and mpMRI were calculated at different cut-off values for the diagnosis of PCa and csPCa. Receiver operating characteristic (ROC) curve analysis was used to assess the diagnostic performance of PSA derivatives for overall PCa and csPCa. The area under the curve (AUC) values were estimated for various PSA derivatives, and a DeLong test using MedCalc (MedCalc Software, Ostend, Belgium) was used to examine the differences in AUCs. All statistical analyses were performed via the IBM SPSS statistics program for Windows, version 23.0 (IBM Corp., Armonk, NY, USA). Statistical significance was determined to be two-sided *p*-values, with *p* < 0.05.

## 3. Results

### 3.1. Demographic and Clinical Characteristics

Table 1 shows the baseline characteristics of the study population. The median (IQR) age was 64.3 (60.0–69.3) years, and the tPSA was 6.64 (4.05–9.54) ng/mL. Prostate volume was 40.2 (28.5–50.5) ml, while PHI was 35.3 (28.6–48.3). Among the 392 patients, 121 (30.9%) were diagnosed with PCa. When patients were categorized on the basis of a diagnosis of PCa, patients with Pca had smaller prostate volumes (36.7 ± 18.0 vs. 47.8 ± 22.6, *p* < 0.001), higher PSA density (PSAD) (0.25 ± 0.19 vs. 0.17 ± 0.24, *p* < 0.001), a lower percentage of fPSA (14.1 ± 6.7 vs. 19.4 ± 8.6, *p* = 0.013), a higher percentage of p2PSA (2.05 ± 1.55 vs. 1.35 ± 1.55, *p* < 0.001), and higher PHI (51.6 ± 19.5 vs. 36.5 ± 15.7, *p* < 0.001) than the patients with negative biopsy reports. The baseline characteristics of patients with PCa are presented in Appendix A. 

### 3.2. Biopsy Results and ROC Curve Analysis for PSA Derivatives

As shown in Figure 2, PCa was identified in 30.9% (121/392) of PI-RADSv2 score-3 lesions, of which 48.8% (59/121) were identified as csPCa. Overall, in terms of PI-RADSv2 score-3 lesions, csPCa was identified in 15.1%. The cancer detection as on-target, off-target, and whole prostate is presented in Appendix A.

When applying ROC analysis for the diagnostic detection of PCa in patients with PI-RADS v2 score-3 lesions, PHI and PSAD yielded a significantly higher AUC of 0.807 (95% CI: 0.710–0.906) and 0.712 (95% CI: 0.611–0.813) for overall PCa compared with PSA (*p* = 0.001 and *p* = 0.016) (Table 2 and Figure 3A). Similarly, for csPCa, PHI and PSAD also yielded a significantly higher AUC of 0.819 (95% CI: 0.723–0.922) and 0.760 (95% CI: 0.652–0.866,) compared with PSA (*p* < 0.001 and *p* = 0.014) (Table 2 and Figure 3B).

### 3.3. Diagnostic Performance According to Different PHI Cut-Offs

Table 3 shows a comparison of diagnostic performance between various PHI cut-off values for overall PCa and csPCa. When biopsy was restricted to those patients with PI-RADSv2 score-3 lesions and a PHI of ≥27, 25.0% of unnecessary biopsies could have been avoided, but 5.0% of PCa could have gone undiagnosed. If biopsy was limited to those patients with PHI ≥ 30, the undiagnosed PCa rate would be 8.3%, but the rate of unnecessary biopsies that could be avoided would be 34.4%. However, there were no cases of missed csPCa diagnosis.

## 4. Discussion

A PSA screening test is generally used for the initial screening of PCa and enables the early detection of PCa. However, in terms of PCa diagnosis with PSA, a number of unnecessary biopsies have been performed because of the method’s characteristics of high sensitivity and low specificity. At the same time, the reported occurrence of clinically insignificant PCa has also increased, which accounts for approximately half of the total occurrences of PCa. This raises concerns among clinicians in terms of overdiagnosis and overtreatment. Consequently, clinicians have the option to use active surveillance (AS) as the go-to treatment option to combat overtreatment and for managing clinically insignificant PCa [20]. However, approximately 15% to 30% of patients with AS experience a GS upgrade shortly after, which suggests misdiagnosis or misclassification at the first biopsy [21,22,23]. In the current era, the importance of identifying cases with csPCa has been widely emphasized. Therefore, various biomarkers have been introduced to diagnose csPCa [24,25].

Since the creation of the PI-RADS system in 2012, mpMRI of the prostate has become a vital imaging tool in the diagnosis of PCa. Most guidelines recommend the implementation of mpMRI in various situations. In clinical practice, mpMRIs have become an important diagnostic tool to help detect suspected lesions. Furthermore, mpMRI reduces the need for performing unnecessary biopsies and enables only targeted biopsies [9,10]. However, with mpMRI, a lesion corresponding to a PI-RADSv2 score-3 lesion is considered to be in the “gray zone” and is categorized as “intermediate,” which means that the lesion has an equivocal risk of presenting as csPCa; clinicians have many concerns about performing a biopsy for these types of lesions.

Recently, many studies have evaluated diagnostic performance in the field by combining PSA derivatives such as tPSA, fPSA, and [-2] proPSA (p2PSA) with mpMRI results. When combined with PI-RADS, PSA derivatives improve diagnostic performance and can be maximized when combined with PHI [26,27,28,29]. One of these studies mentions the value of PHI for PI-RADSv2 score-3 lesions, but the validity was not confirmed because of the small cohort number [25]. The current study is different from previous studies in that we have investigated diagnostic performance by applying PSA derivatives only to PI-RADS v2 score-3 lesions.

To the best of our knowledge, this study is the first to evaluate the use of various PSA derivatives, combined with PI-RADS v2 score-3 lesions, to predict PCa as well as csPCa. In our study of 392 patients, PCa diagnosis was confirmed in 30.9% of patients, and csPCa diagnosis was confirmed in 15.1%. We found that PI-RADS score-3 lesions with PHI had the highest AUC overall in PCa (0.807, 95% CI: 0.710–0.906) and csPCa (0.819, 95% CI: 0.723–0.922) compared with other PSA derivatives with PI-RADS v2 score-3 lesions and could avoid unnecessary biopsies, to some extent, but this included all csPCa.

The results of the overall detection rates of PCa and csPCa for PI-RADSv2 score-3 lesions vary; however, the variation between studies is not that large. Recently, according to a meta-analysis, the detection rate of overall PCa was 33% (95% CI, 27–39) and for csPCa it was 17% (95% CI, 13–21) for PI-RADSv2 score-3 lesions [30]. In a prospective study by Krüger-Stokke et al., the detection rate for overall PCa is 37%, and for csPCa is 21% on PI-RADSv2 score-3 lesions [31]. Our study shows that the overall PCa detection rate is 30.9%, and csPCa detection rate is 15.1% for PI-RADSv2 score-3 lesions. Our results are comparable with those of other studies, which supports the reliability of our research.

To date, several studies have examined ways to reduce unnecessary biopsies and improve the diagnosis of PCa and csPCa on PI-RADSv2 score-3 lesions. A number of studies have reported that PSAD is associated with biopsy results in PI-RADSv2 score-3 lesions. According to a study by Natale et al., PSAD is a useful tool that can be consulted to determine whether a biopsy in patients with PI-RADS3 is required. [32]. Zhang et al. also reported that PSAD could improve the detection of PCa, as well as csPCa, for PI-RADSv2 score-3 lesions [33]. Rico et al. reported that PSAD improves the specificity and PPV, contributing to the improved management of csPCa [34]. In addition, there is a study showing that, along with PSAD, age could be a way to avoid excessive biopsies [35]. Awamlh et al. reported that a higher PSAD, a lower prostate volume, and lower apparent diffusion coefficient (ADC) values are related to csPCa [36]. In our study, compared with PSA, the PSAD and PHI yielded significantly higher AUCs by which to diagnose PCa, as well as csPCa, for PI-RADSv2 score-3 lesions. These results indicated that similar to PSAD, PHI also improved the diagnostic performance of targeted cores in demonstrating csPCa in PI-RADSv2 score-3 lesions of interest. However, as the 95% CIs of PHI and PSAD clearly overlap, they did not show a significant difference in diagnostic performance. 

Despite the clinical implications of our study, there are certain limitations. First, owing to the study’s retrospective design, an inherent structural bias may exist. Second, our cohort cannot represent a general population but rather represents a population with elevated PSA or people who are concerned about their physical status, thus raising concerns of selection bias. Third, although our study is the largest one so far to evaluate the clinical use of PHI with PI-RADSv2 score-3 lesions, the number of cohorts is relatively limited, and the study would be better with a greater number of cohorts. Fourth, the biopsy methods might affect the results. As we only examined cognitive-targeted biopsies by experienced uro-radiologists, applying other biopsy techniques might yield results different from those of our study. Fifth, in our study, we utilized TRUS to assess prostate volume, although another study has shown that this particular method might be relatively inaccurate [37]. In fact, when prostate volume was measured with an ellipsoid formula, the median prostate volume and PSAD were 40.2 mL and 0.16, respectively. However, when prostate volume was measured using MRI planimetry, the median prostate volume and PSAD were 35.2 mL and 0.19, respectively. These discrepancies could influence the diagnostic performance of PSAD [38]. Finally, our results could not fully represent the patients’ cancer status due to the heterogeneity of PCa. It could be possible that biopsies could have missed or understated PCa. 

## 5. Conclusions

Introducing PHI or PSAD in the decision process to examine the targeted cores of PI-RADSv2 score-3 lesions of interest showed higher accuracy in the diagnosis of PCa, as well as csPCa, compared with a combination of other PSA derivatives. If biopsies were restricted to patients with a PI-RADSv2 score of 3 and a PHI of ≥30, then up to 34.4% of prostate biopsies could be avoided. In addition, there were no cases of missed csPCa diagnosis. Factoring in both the PHI and PI-RADSv2 score-3 lesions is expected to contribute majorly to diagnosing PCa, as well as csPCa, and to the decision regarding biopsy in the clinical setting. Confirmation regarding integrating PHI and mpMRI to use in the diagnosis of PCa and csPCa via external validation studies is necessary. 

## Figures and Tables

**Figure 1 cancers-14-04174-f001:**
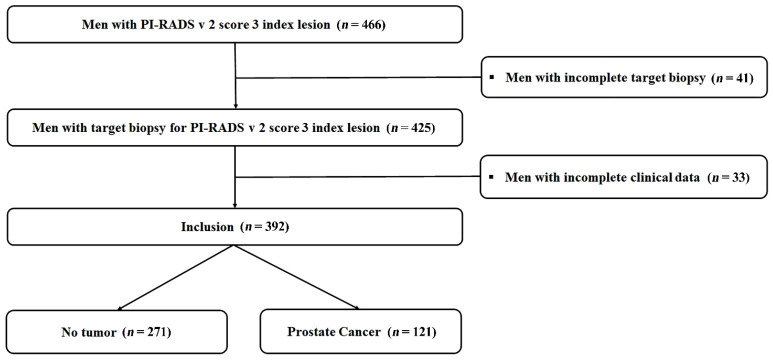
Flowchart of the study.

**Figure 2 cancers-14-04174-f002:**
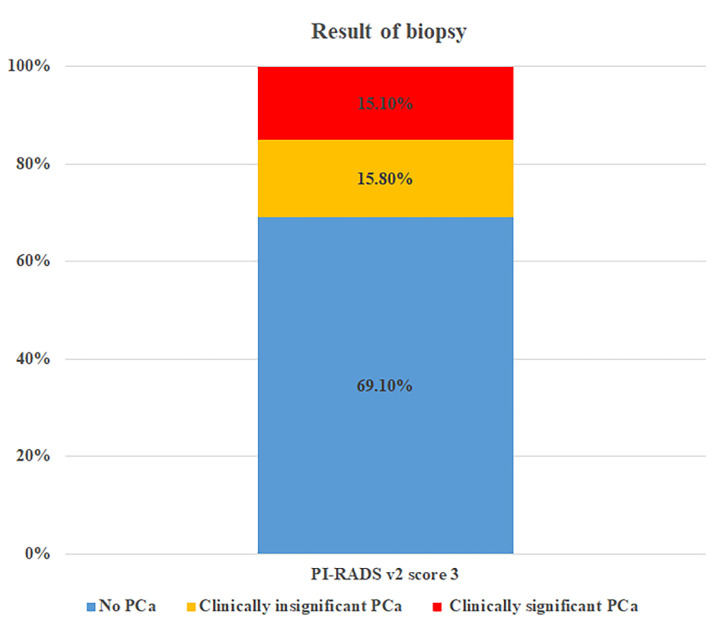
Pathologic outcomes of PI-RADS v2 score-3 lesions, as assessed by cognitive, TRUS-targeted cores.

**Figure 3 cancers-14-04174-f003:**
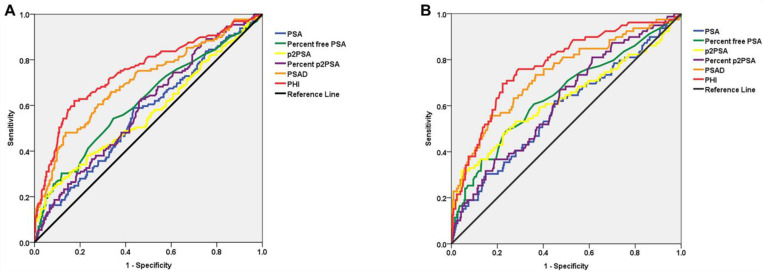
Receiver operating characteristic (ROC) curve analysis for PSA derivatives for the prediction of (**A**) overall prostate cancer and (**B**) clinically significant prostate cancer.

**Table 1 cancers-14-04174-t001:** Baseline characteristics.

Variables	Total	Detection of Prostate Cancer		*p*
		No	Yes	
No. of Patients, *n* (%)	392 (100.0)	271 (69.1)	121 (30.9)	
Age, years				0.725
Median (IQR)	64.3 (60.0–69.3)	64.0 (60.0–69.0)	65.0 (60.0–70.0)	
Mean (SD)	63.9 (8.2)	63.7 (8.7)	64.4 (7.3)	
Total PSA, ng/ml				0.289
Median (IQR)	6.64 (4.05–9.54)	6.56 (4.17–9.85)	6.85 (4.31–10.18)	
Mean (SD)	8.54 (8.96)	8.21 (7.20)	9.31 (9.21)	
Prostate volume, ml				<0.001
Median (IQR)	40.2 (28.5–50.5)	42.6 (31.3–52.2)	34.6 (23.9–44.1)	
Mean (SD)	44.5 (21.6)	47.8 (22.6)	36.7 (18.0)	
PSAD				<0.001
Median (IQR)	0.16 (0.10–0.26)	0.14 (0.11–0.28)	0.20 (0.10–0.23)	
Mean (SD)	0.19 (0.23)	0.17 (0.24)	0.25 (0.19)	
% fPSA				0.013
Median (IQR)	16.8 (12.2–23.6)	19.6 (13.9–24.8)	11.5 (9.9–18.3)	
Mean (SD)	17.9 (7.5)	19.4 (8.6)	14.1 (6.7)	
p2PSA, pg/ml				<0.001
Median (IQR)	18.35 (15.59–30.13)	15.46 (11.28–22.25)	24.24 (18.62–38.74)	
Mean (SD)	19.69 (8.16)	16.04 (8.46)	27.62 (7.49)	
% p2PSA				<0.001
Median (IQR)	1.42 (1.27–2.01)	1.25 (0.99–1.52)	1.81 (1.49–2.23)	
Mean (SD)	1.68 (1.56)	1.35 (1.55)	2.05 (1.55)	
PHI				<0.001
Median (IQR)	35.3 (28.6–48.3)	30.5 (26.0–44.3)	46.5 (35.0–62.1)	
Mean (SD)	39.2 (17.3)	36.5 (15.7)	51.6 (19.5)	
Biopsy Gleason grade, *n* (%)				
6			62 (51.2)	
7 (3 + 4)			47 (38.9)	
8			12 (9.9)	

**Table 2 cancers-14-04174-t002:** Comparison of the AUC of PSA derivatives, based on total PSA in patients with PI-RADS v2 score-3 lesions, for the detection of overall and clinically significant prostate cancer.

Variables	Overall Prostate Cancer			Clinically Significant Prostate Cancer		
	AUC	95% CI	*p*	AUC	95% CI	*p*
Total PSA	0.593	0.481–0.688	Ref	0.604	0.502–0.714	Ref
PSAD	0.712	0.611–0.813	0.016	0.760	0.652–0.866	0.014
% fPSA	0.661	0.566–0.772	0.231	0.682	0.589–0.772	0.248
p2PSA	0.613	0.521–0.704	0.304	0.662	0.554–0.756	0.326
% p2PSA	0.630	0.542–0.744	0.135	0.648	0.539–0.742	0.216
PHI	0.807	0.710–0.906	0.001	0.819	0.723–0.922	<0.001

**Table 3 cancers-14-04174-t003:** Diagnostic performance for overall and clinically significant prostate cancer detection via different PHI cutoffs.

	Sensitivity	Specificity	PPV	NPV	Biopsy Criteria by PHI	
					% Biopsy Avoided	% PCa Missed
PHI ≥ 27	95.0%	33.9%	39.1%	93.9%	25.0%	5.0%
PHI ≥ 30	91.7%	46.1%	43.2%	92.6%	34.4%	8.3%
PHI ≥ 33	86.0%	56.5%	46.8%	90.0%	43.4%	14.0%
PHI ≥ 36	75.2%	62.7%	47.3%	85.0%	51.0%	24.8%
PHI ≥ 40	66.1%	70.1%	49.7%	82.3%	58.7%	33.9%

## Data Availability

The dataset used and/or analyzed during the current study is available from the corresponding author upon reasonable request.

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
