# Peer review of "Clinical Utility of Prostate Health Index for Diagnosis of Prostate Cancer in Patients with PI-RADS 3 Lesions"

_cancers, 2022, doi:10.3390/cancers14174174_

Round 1
Reviewer 1 Report
The authors of this study investigate the clinical relevance of Prostate Health Index for diagnosis of prostate cancer in patients with PI-RADS 3 lesions.
While this is an interesting work with potential value, the following issues need to be addressed:
- The word "use" in the title does not reflect utility, maybe rephrasing to utility or value would be more appropriate
- what is the cost of calculating the PHI relatively to PSA per patient per test?
- the authors should try to provide external validation of PHI in a different cohort as this is a major limitation of the study, along with its retrospective nature
- did the authors collect any additional clinical information re: inflammation markers, recent UTI etc on the patients who were assessed for PHI?
- How many of the patients that initially screened negative for cancer ended up with a prostate cancer diagnosis on serial assessments? This information is not disclosed.
-
Author Response
We appreciate the very detailed comments that you gave us on our study. In light of your feedback, we realized the limitations of our paper that we did not previously recognize, and through this, we sought to improve our study and ultimately our paper as well. Although our work has irreparable limitations, we expect our work to contribute in part to the development of prostate cancer research.
Comments and Suggestions for Authors
The authors of this study investigate the clinical relevance of Prostate Health Index for diagnosis of prostate cancer in patients with PI-RADS 3 lesions.
While this is an interesting work with potential value, the following issues need to be addressed:
1) The word "use" in the title does not reflect utility, maybe rephrasing to utility or value would be more appropriate
We agree with your opinion. We changed “use” to “utility”.
2) What is the cost of calculating the PHI relatively to PSA per patient per test?
The cost of PSA is about $11.5, and PHI costs around $150 in South Korea.
3) The authors should try to provide external validation of PHI in a different cohort as this is a major limitation of the study, along with its retrospective nature
We agree with your opinion. External validation is a necessary process to supplement the limitation of our study. However, due to time limitations, we could not present data of external validation. Instead, we plan to conduct a multi- center study based on our data. We will report it as a large study later.
4) Did the authors collect any additional clinical information re: inflammation markers, recent UTI etc on the patients who were assessed for PHI?
We did not routinely check inflammation markers. Instead, urine analysis with microscopy was routinely checked. For patients with UTI, PSA was re-checked after sufficient antibiotic use. The value of PSA and PHI are obtained when there is no sign of inflammation.
5) How many of the patients that initially screened negative for cancer ended up with a prostate cancer diagnosis on serial assessments? This information is not disclosed.
In our institutional protocol, if prostate cancer is not confirmed by the target biopsy but prostate cancer is still clinically suspected, we usually recommend and perform transperineal biopsy. Therefore, there are limitations to suggest the detection rate of prostate cancer through serial assessment in the PI-RADS score 3 lesion. We ask for your understanding for these limitations.

Reviewer 2 Report
General comment
The manuscript entitled “Clinical Use of Prostate Health Index for Diagnosis of Prostate Cancer in Patients with PIRADS 3 lesions” aims to evaluate the role of PHI in the diagnosis of prostate cancer in patients with ambiguous PIRADS 3 lesions at the mpMRI. The manuscript is overall well written and the argument it is quite interesting and on point. A few corrections are required in order to improve the quality of the work before considering the manuscript suitable for the publication.
- Major issues
Introduction
47-51: The introduction is a bit rushed and does not take into consideration the epidemiology of prostate cancer and the development and the advantages related to PHI
Materials and Methods
69: Add inclusion and exclusion criteria
104-110: Regarding the biopsy protocol, it is not clear which were the criteria for doing the systemic versus the targeted biopsy. Please clarify.
Results
138: How and when the prostate volume was assessed? Please report this in the methods and, to this regard, see also DOI: 10.1159/000516681 and DOI: 10.1007/s11255-016-1350-8
Discussion
232-242: In the limitations, another that has to be added is related to the heterogeneity of prostate cancer. As result, it could be possible that biopsies could have missed or understaged PCa. A similar limitation is related to the absence of a histopathological confirmation of the postoperative specimen which, albeit it is beyond the aim of this study, could further improve the reliability of PHI.
- Minor issues
Materials and methods
119: how did you assess the normality of data before proceeding to the statistical analysis?
Conclusions
Refer, briefly, to future perspectives
References
268: Update the references to 2010 at least, when possible.
Author Response
We appreciate the very detailed comments that you gave us on our study. In light of your feedback, we realized the limitations of our paper that we did not previously recognize, and through this, we sought to improve our study and ultimately our paper as well. Although our work has irreparable limitations, we expect our work to contribute in part to the development of prostate cancer research.
General comment
The manuscript entitled “Clinical Use of Prostate Health Index for Diagnosis of Prostate Cancer in Patients with PIRADS 3 lesions” aims to evaluate the role of PHI in the diagnosis of prostate cancer in patients with ambiguous PIRADS 3 lesions at the mpMRI. The manuscript is overall well written and the argument it is quite interesting and on point. A few corrections are required in order to improve the quality of the work before considering the manuscript suitable for the publication.
- Major issues
Introduction
47-51: The introduction is a bit rushed and does not take into consideration the epidemiology of prostate cancer and the development and the advantages related to PHI
For your recommendation, we added the information about the epidemiology of prostate cancer, and development and the advantages related to PHI. Please see the revised manuscripts page 1-2 line 42-43 and 49-56.
Materials and Methods
69: Add inclusion and exclusion criteria
Please see the revised manuscripts page 2 line 83-89 starting with the sentence “We retrospectively reviewed 466 patients whose…”
104-110: Regarding the biopsy protocol, it is not clear which were the criteria for doing the systemic versus the targeted biopsy. Please clarify.
Initially, TRUS-guided, cognitive-targeted biopsy with more than one core of sample in PI-RADS v2 3 lesions was performed. Then systemic, 12-core specimens were obtained thereafter. Please see the revised manuscripts page 3, 2.4 Biopsy protocol, line 112-121
Results
138: How and when the prostate volume was assessed? Please report this in the methods and, to this regard, see also DOI: 10.1159/000516681 and DOI: 10.1007/s11255-016-1350-8
We added assessment of prostate volume at 2. Materials and Methods – 2.4 Biopsy protocol. Please see the revised manuscripts page 3 line 114-116 starting with the sentence “Before performing prostate biopsy, prostate volume was ….”
Discussion
232-242: In the limitations, another that has to be added is related to the heterogeneity of prostate cancer. As result, it could be possible that biopsies could have missed or understaged PCa. A similar limitation is related to the absence of a histopathological confirmation of the postoperative specimen which, albeit it is beyond the aim of this study, could further improve the reliability of PHI.
We fully agree with your opinion. Therefore, the above mentioned contents have been added to the limitation section. Please see the revised manuscripts page 8 line 251-254 starting with the sentence “Finally, our results could not fully ….”
- Minor issues
Materials and methods
119: how did you assess the normality of data before proceeding to the statistical analysis?
The normality of the original data was evaluated using the Kolmogorov-Smirnov test
Conclusions
Refer, briefly, to future perspectives
We added future perspectives in conclusion section. Please see the revised manuscripts page 8 line 261-264 starting with the sentence “The combination of PHI and PI-RADS v2 score 3 lesions ….”
References
268: Update the references to 2010 at least, when possible.
As your recommendation, we updated all references to 2010 at least.

Reviewer 3 Report
It was a pleasure to read this useful, well-written paper. Its main finding is that - in patients with PI-RADS v2 score 3 lesions - using 30 as a threshold for PHI could help to avoid every third (unnecessary) biopsy. Notably, 8.3% of cis PCsa, and none csPCa was missed by that threshold. This conclusion is of clinical importance. On the other hand, such a low threshold is likely to fuel some debate. The Methods section contains all the necessary information. The study design is appropriate, well explained and illustrated with a flowchart. The presentation of the results is logical and comprehensive. The limitations are fairly addressed. I congratulate the authors on this excellent work. I only suggest some minor improvements. 1) The authors examined the diagnostic accuracy of PHI up to the threshold of 36 (Table 3). However, a generally accepted threshold for PHI is usually 40 or 41. It would be advisable to include the results obtained at PHI >40 in the table. 2) It is worth noting that the calculations were performed for different thresholds, providing differentiated and useful data to physicians and well-informed patients. However, a calculated "optimal decision threshold" or "Youden index" could complement the presentation of the diagnostic performance of the pHI in the studied cohort. 3) Introduction, line 46/47: "Several diagnostic tools have been developed to improve the detection of PCa and studied for their clinical efficacy". In 2-4 sentences, please provide some examples of these improvements (e.g." free-to-total PSA ratio", "PSA velocity", etc.) and indicate their limitations. It would better substantiate the focus on PHI in the present study. 4) There exist several definitions of clinical significant/insignificant PCa, e.g. EAU /ISUP criteria (used in the present study; grade group > 2), but also Epstein criteria etc. Please, explain the therapeutic consequences of cs/cis for readers less familiar with urological oncology. 5) Introduction, line 77: typo error "cognitive, target biopsy". It should be "cognitive-targeted biopsy") 6) Methods: "MediCalc" - do the authors mean the "Medical Calculatior System" or rather the "MedCalc" statistic software? Please add a corresponding reference or manufacturer.Author Response
We appreciate the very detailed comments that you gave us on our study. In light of your feedback, we realized the limitations of our paper that we did not previously recognize, and through this, we sought to improve our study and ultimately our paper as well. Although our work has irreparable limitations, we expect our work to contribute in part to the development of prostate cancer research.
Reviewer 3
It was a pleasure to read this useful, well-written paper. Its main finding is that - in patients with PI-RADS v2 score 3 lesions - using 30 as a threshold for PHI could help to avoid every third (unnecessary) biopsy. Notably, 8.3% of cis PCa, and none csPCa was missed by that threshold. This conclusion is of clinical importance. On the other hand, such a low threshold is likely to fuel some debate. The Methods section contains all the necessary information. The study design is appropriate, well explained and illustrated with a flowchart. The presentation of the results is logical and comprehensive. The limitations are fairly addressed. I congratulate the authors on this excellent work. I only suggest some minor improvements.
1) The authors examined the diagnostic accuracy of PHI up to the threshold of 36 (Table 3). However, a generally accepted threshold for PHI is usually 40 or 41. It would be advisable to include the results obtained at PHI >40 in the table.
As your recommendation, we added the results obtained at PHI ≥40. Please revised Table 3 in page 6.
2) It is worth noting that the calculations were performed for different thresholds, providing differentiated and useful data to physicians and well-informed patients. However, a calculated "optimal decision threshold" or "Youden index" could complement the presentation of the diagnostic performance of the pHI in the studied cohort.
A calculated Youden Index was PHI≥41.5 (sensitivity 0.62, specificity 0.81) for overall PCa. For csPCa, a calculated Youden Index was PHI≥35 (sensitivity 0.76, specificity 0.72)
3) Introduction, line 46/47: "Several diagnostic tools have been developed to improve the detection of PCa and studied for their clinical efficacy". In 2-4 sentences, please provide some examples of these improvements (e.g." free-to-total PSA ratio", "PSA velocity", etc.) and indicate their limitations. It would better substantiate the focus on PHI in the present study.
As your recommendation, we mentioned several diagnostic tools and their limitations. Please see the revised manuscripts page 2 line 49-53 starting with the sentence “For example, free PSA (fPSA), free-to-total PSA ratio ….”
4) There exist several definitions of clinical significant/insignificant PCa, e.g. EAU /ISUP criteria (used in the present study; grade group > 2), but also Epstein criteria etc. Please, explain the therapeutic consequences of cs/cis for readers less familiar with urological oncology.
As your recommendation, we added additional explanation in Introduction. Please see the revised manuscripts page 2 line 56-59 starting with the sentence “Although csPCa requires a definite treatment such as ….”
5) Introduction, line 77: typo error "cognitive, target biopsy". It should be "cognitive-targeted biopsy")
As your recommendation, we changed to “cognitive-targeted biopsy”. (Revised manuscripts page 2 line 86)
6) Methods: "MediCalc" - do the authors mean the "Medical Calculatior System" or rather the "MedCalc" statistic software? Please add a corresponding reference or manufacturer.
We changed to “MedCalc” and add a manufacturer. (Revised manuscripts page 4 line 141)

Round 2
Reviewer 1 Report
No additional comments.
Author Response
We appreciate the very detailed comments that you gave us on our study. We could improve our study and ultimately our paper as well. Thank you again.
Reviewer 2 Report
The authors improved the quality of the manuscript according to previous suggestions. Considering that the volume of the prostate was obtained via TRUS, it should be reported, among minor limitation of the study, the relative inaccuracy of this measuring methods, according to the references previously suggested.
(138: How and when the prostate volume was assessed? Please report this in the methods and, to this regard, see also DOI: 10.1159/000516681 and DOI: 10.1007/s11255-016-1350-8)
Author Response
We agreed with your opinion. Prostate volume assessed by TRUS might be relatively inaccurate. We added sentence in limitation part starting with "Fifth, in our study, we utilized TRUS to ..."
